# Neural Network-Based Solar Irradiance Forecast for Edge Computing Devices

**Georgios Venitourakis** [1,†] , **Christoforos Vasilakis** [1,†] , **Alexandros Tsagkaropoulos** [1,†] , **Tzouma Amrou** [2,†] , **Georgios Konstantoulakis** [2,†] , **Panagiotis Golemis** [2,†] and **Dionysios Reisis** [1,*,†]

1   Electronics Lab, Physics Department, National & Kapodistrian University of Athens, 15772 Athens, Greece; gvenit@phys.uoa.gr (G.V.); christof-v@phys.uoa.gr (C.V.); atsagk@phys.uoa.gr (A.T.)
2   Inaccess Networks, 12 Sorou Str., 15125 Maroussi, Greece; tamrou@inaccess.com (T.A.); gkonst@inaccess.com (G.K.); golemispan@inaccess.com (P.G.)
*   Correspondence: dreisis@phys.uoa.gr; Tel.: +30-210-727-6708
†   These authors contributed equally to this work.

**Abstract:** Aiming at effectively improving photovoltaic (PV) park operation and the stability of the electricity grid, the current paper addresses the design and development of a novel system achieving the short-term irradiance forecasting for the PV park area, which is the key factor for controlling the variations in the PV power production. First, it introduces the Xception long short-term memory (XceptionLSTM) cell tailored for recurrent neural networks (RNN). Second, it presents the novel irradiance forecasting model that consists of a sequence-to-sequence image regression NNs in the form of a spatio-temporal encoder–decoder including Xception layers in the spatial encoder, the novel XceptionLSTM in the temporal encoder and decoder and a multilayer perceptron in the spatial decoder. The proposed model achieves a forecast skill of 16.57% for a horizon of 5 min when compared to the persistence model. Moreover, the proposed model is designed for execution on edge computing devices and the real-time application of the inference on the Raspberry Pi 4 Model B 8 GB and the Raspberry Pi Zero 2W validates the results.

**Keywords:** deep learning; ConvLSTM; irradiance forecasting; edge computing; photovoltaic parks; ground-based sky images

## 1. Introduction

Artificial intelligence (AI) is an ever-expanding technology that has spread in unconventional scientific and industrial fields and has been integrated in smart systems [1–3] in order to execute notorious tasks such as predicting the future state of a system and decision making. These tasks often appear in the Smart Grid (SG) concept [3,4], where power grids are supported by the information provided by Internet of Things (IoT) devices, which are constantly monitoring the environment and the interaction between the energy provider and the client. SGs are essential to power production involving renewable energy sources (RES) because the RES and especially the photovoltaic (PV) parks have the disadvantage of not producing energy at a constant rate.

In PV parks, the energy production depends heavily on the global horizontal irradiance (GHI), the diffuse horizontal irradiance (DHI), the direct normal irradiance (DNI), the cloud cover (CC) and other meteorological parameters. AI-enabled smart PV parks utilize machine learning (ML) tools to forecast the future values of these parameters [5]. These forecasting results lead the PV park controller to improve the power production capabilities and henceforth the grid balancing [6]. The problem in many PV parks is the lack of historical data that prevents a neural network (NN) from making reliable predictions. It is also the case that numerical values are often not sufficient to depict the current state of a system such as the weather conditions in the atmosphere. For this reason, researchers and engineers considered sky images taken from the ground as an attractive solution. This

is because sky images carry significantly more information compared to numerical data, and moreover, they are able to provide even further detailed information by the use of advanced ML techniques and convolutional NN (CNN)-based models. In the last decade, the approaches with image-regression-based techniques have shown improved results and these advances have made edge computing applications possible by the use of lightweight models such as ShuffleNet [7] and the MobileNet [8].

The motivation of this research is the Archon project [9] that designs and develops a controller that is efficient with respect to the implementation cost, the computational load and the deployment, which will manage the PV park power production [9]. The controller refers to all the aspects of the controlling mechanism, ranging from the underlying infrastructure to the controller's software. The infrastructure includes the sensors that generate data and the hardware solutions for deploying the irradiance forecasting system (IFS) on the edge. The cost-efficient controller design excludes sensors that generate numeric data in order to reduce the cost of the equipment and the complexity of the interconnection of all the contributing devices in the system. For the same purpose, the design excluded the pyranometer, a device that provides essential information regarding current weather conditions, because it is in high demand and expensive. Therefore, the entire design of the proposed IFS depends solely on image sequences captured by low-cost cameras. Moreover, the implementation targets edge computing devices that require less energy and are of lesser cost compared to a work station.

Aiming at an improved solution to the GHI forecasting problem executable on edge devices, the current article exploits image sequence regression techniques to introduce the following two novel entities: (a) the Xception long short-term memory (XceptionLSTM), a recurrent NN (RNN) for image sequence parsing and generation, and (b) a model for a complete solution to the GHI forecasting that uses the proposed XceptionLSTM to improve the forecasting results. The design of the proposed XceptionLSTM cell is based on convolutional LSTM (ConvLSTM) cells. The proposed GHI forecasting model is a sequence-to-sequence (Seq2Seq) image regression NN in the form of a spatio-temporal encoder–decoder [10] that consists of Xception layers (XL) [11] in the spatial encoder, the novel XceptionLSTM cells proposed by this work in the temporal encoder and decoder and a multilayer perceptron (MLP) in the spatial decoder. The novel XceptionLSTM has the following advantages compared to the ConvLSTMs: (a) its design allows the parallelized execution of ConvLSTM cells with different kernel sizes, (b) it is significantly more lightweight and (c) it showcases significantly improved usage of the data and kernel tensors. The novel GHI forecasting model has the following improvements with respect to the reported Seq2Seq architectures based on ConvLSTMs: (a) the proposed model converges faster and (b) it requires less memory in order to infer data, making it ideal for executing inference on-the edge devices.

The development and evaluation of the proposed model employs a custom dataset of red green blue (RGB) $180°$ field-of-view sky images and GHI measurements collected over a full callendar year period during the development of the the Archon project [9]. The proposed model is trained and evaluated on an NVIDIA GeForce RTX 3080. The design of the entire model has as target the execution on edge computing devices. For the time performance evaluation we opted the Raspberry Pi 4 Model B 8 GB and the Raspberry Pi Zero 2W as low-power, edge computing devices. The development of the model was based on Pytorch [12], which is a Python package and a framework for NNs with a relatively high Graphics Processing Unit (GPU) acceleration. Pvlib [13] is a Python package for PV park performance simulation; in this work, it was the basis for estimating the position of the sun in an image and for the generation of sun masks [14].

The paper is organised as follows. First, Section 2 introduces the XceptionLSTM cell and the model for the short-term irradiance forecasting. Section 3 reports the evaluation results of CNN models for irradiance forecasting. Section 4 follows with a discussion regarding the results presented in this and other related work reported in the literature. Finally, Section 5 concludes the article.

## 2. Materials and Methods

Making decisions proactively for a system that we monitor relies on the following assumption: the system's next state depends on the sequence of states in the recent history of that system. Accordingly, we need the accurate forecasts for the forthcoming GHI state in order to improve the PV park's operation. In order to accomplish the latter task, a sky camera captures a sequence of images consecutively with a constant time interval (referred to as horizon from now on). Then, the PV park's controller forwards the images to an NN. The prediction model calculates a new sequence of values that corresponds to consecutive GHI values with the same horizon as the input sequence. The input and output sequence length and the horizon used for the prediction are the major model's hyperparameters that can be tuned to achieve the most accurate possible outcome. Other hyperparameters are the model's structure, the training schemes and any data preprocessing.

### 2.1. Model Structure

Most often, NNs need to be quite complex because of the large systems they are tasked to simulate. Therefore, in highly complex systems such as the Earth's atmosphere, the traditional models that can respond to such fast-changing parameters may consist of dozens of sequentially connected layers. Such models are most probably time consuming in the tasks of training and inference and they are considered non-optimal for time-critical applications and on-the-edge inference. These facts show that such systems require efficient state-of-the-art ML algorithms and novel techniques with improved complexity and lesser requirements for computational resources.

#### 2.1.1. Xception Layer

The proposed prediction model utilizes XLs [11], which are a type of CNN that combines the characteristics of Inception Modules [15] and depthwise separable convolutions (DWSC) [11,16]. A DWSC extracts the parallelism of a traditional convolutional layer (CL) by partitioning the operation in two simpler operations: a depthwise convolution and a pointwise convolution. The former is a convolution in each frame of the channels of the input tensor, while the latter is a convolution in each pixel of the input tensor. Combining the depthwise and pointwise convolution sequentially results in a CL with the same result-producing capabilities but with a significantly reduced number of parameters and computational complexity. The computational graph of a DWSC is depicted in Figure 1. An Inception Module consists of a nested CL, where all nested layers process the same input in parallel, and all the results are concatenated, added or in general reduced to a new output tensor. The layers' parallelization leads to a greater degree of data usage, making it an ideal layer for inference on low-power edge devices. The benefit of employing Inception Modules is also the attenuation of the vanishing gradient problem [17]. This problem refers to those models that have a great number of sequentially connected layers, and during the training scheme in these models, the gradient often becomes insignificant during backpropagation.

The combination of DWSC and Inception Modules results in the XLs, which execute in parallel depthwise operations such as the depthwise convolutions or the pooling operations. Then, the model concatenates these results and forwards them to a pointwise convolution. The latter scheme exploits two forms of parallelism: the inter-task parallelism (parallel execution of nested layers in the Inception Module) and intra-task parallelism (parallel execution of convolutions in all channels of the input tensor in a depthwise convolution). Although the CL allows for an intra-task parallelism because of the different kernels that can be applied independently in parallel (called inter-kernel parallelism), the proposed model's design can achieve a greater degree of intra-task parallelism with an XL and the depthwise convolutions; the latter allow the partition of the input tensor into the channels of the tensor. Hence, we can operate on each of these channels independently. The advantage of the XL compared to the traditional CL is less overhead in terms of memory access and smaller number of parameters and operations. We can add to the above a further

improvement. This is the partitioning of the depthwise convolution in two asymmetrical depthwise convolutions, one for each dimension of the frame, meaning convolutions with kernel size $1 \times N$ and $N \times 1$ [18], where $N \times N$ is the kernel size of the traditional CL. This fact allows even further parallelism of the computations.

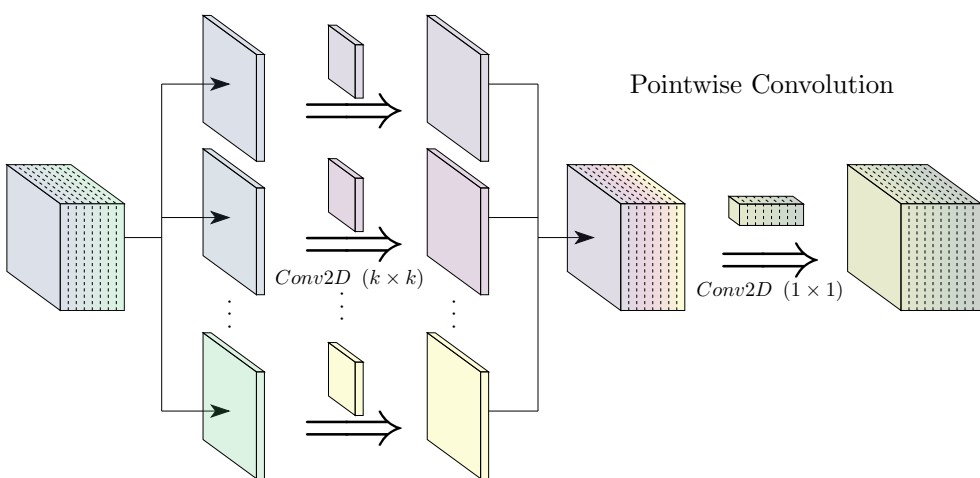

**Figure 1.** Depthwise separable convolution breakdown.

This work considers the structure of the XL shown in Figure 2. It consists of four nested layers, two of which are depthwise convolutions with kernel sizes of 3 and 5, a max pooling layer and the identity function. The identity function is ultimately used as a pointwise convolution of the XL's input tensor. Also, including the identity function improves the gradient descent; the gradient is broadcasted to and propagates through all four nested layers, but is unaffected by the identity function. The gradients of all the nested layers are then added and (back)propagated to the previous layer. As a result, the gradient of an XL is mostly affected by the pointwise convolution and less affected by the nested depthwise convolutions (they can be viewed as small adjustments in the output gradient). Thus, the layers in the later stages of the backpropagation are less likely to experience the vanishing gradient descent problem [15]. We can optionally add an activation function between the depthwise and pointwise operations. However, early results have shown that an activation before the pointwise convolution diminishes the accuracy of the results.

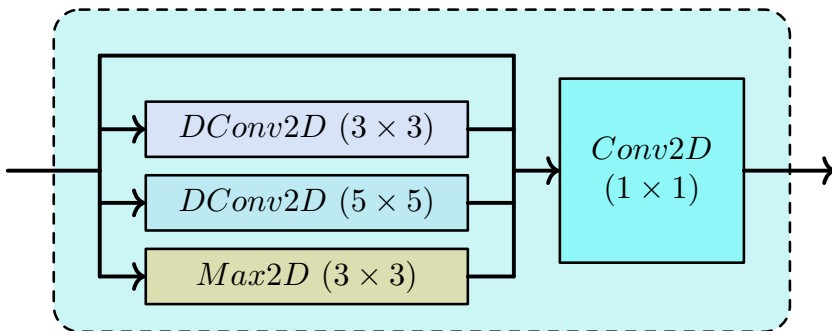

**Figure 2.** Structure of the implemented Xception Layer (XL).

### 2.1.2. XceptionLSTM

ConvLSTM cells [19] are RNNs that utilize convolutions and operate on tensors, in contrast to fully-connected LSTM (FC-LSTM) cells [20] that operate on vectors. The use of LSTM cells is either to process or generate a sequence of data; their most usual applications are the deep learning (DL) based natural language processing (NLP) applications [21]

and time-series predictions [22,23]. ConvLSTM cells are able to process more complex tasks, such as next frame prediction [24] and other time-series predictions with feature extraction [25]. The proposed model utilises XLs [11] as a substitute for convolutions in ConvLSTM. XLs used in the proposed model's LSTM cells are structured as shown in Figure 3. Equations (1)–(10) and Figure 3 describe XceptionLSTM cells:

$$x = \text{Concat}(X_t, H_{t-1}), \tag{1}$$

$$i_c = \text{XL}_i(x), \tag{2}$$

$$f_c = \text{XL}_f(x), \tag{3}$$

$$c_c = \text{XL}_c(x), \tag{4}$$

$$o_c = \text{XL}_o(x), \tag{5}$$

$$i_g = \sigma(i_c + C_{t-1} \odot W_{hi}), \tag{6}$$

$$f_g = \sigma\left(f_c + C_{t-1} \odot W_{hf}\right), \tag{7}$$

$$o_g = \sigma(o_c + C_{t-1} \odot W_{ho}), \tag{8}$$

$$C_t = f_g \odot C_{t-1} + i_g \odot \text{act}(c_c), \tag{9}$$

$$H_t = o_g \odot \text{act}(C_t), \tag{10}$$

where the new input and the hidden state of the previous iteration of the cell are concatenated (Equation (1)) and forwarded to four XLs (input, forget, cell and output convolutions, Equations (2)–(5)). The input, forget and output gates (Equations (6)–(8)) are calculated by first summing the results of the corresponding convolutions with the Hadamard products of the cell state of the previous iteration ($C_{t-1}$) with their corresponding parameters and then applying the sigmoid function. The new cell state ($C_t$) is a combination of the previous cell state, the input and forget gate and the cell convolution (Equation (9)). The new hidden state ($H_t$) is the Hadamard product of the output state with the activated new cell state (Equation (10)). We note here that the XL operations in the XceptionLSTM cell do not have a nested max pooling operation.

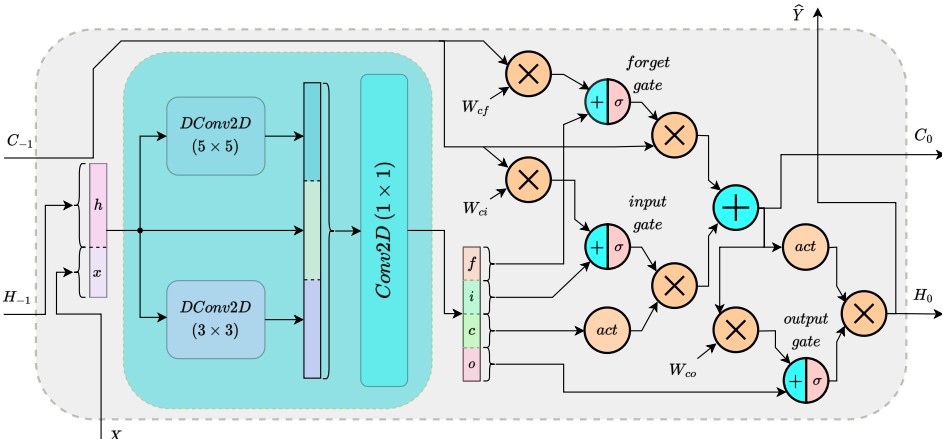

**Figure 3.** The structure of an Xception long short-term memory (XceptionLSTM) cell.

The proposed RNN has significant advantages compared to ConvLSTMs:

- Parallelized execution of multiple ConvLSTM cells with different kernel sizes in a single XceptionLSTM cell.
- Significantly more lightweight when compared to the ConvLSTMs that have similar structural elements.
- Improved utilization of the data and kernel tensors: $k$ times less input data calls in the depthwise convolution and $w^2$ times less kernel calls in the pointwise convolution when compared to traditional convolutions, where $k$ is the number of kernels and $w$ is the window size of the tradithonal CL.

### 2.1.3. Proposed Model

The proposed model is a spatio-temporal encoder/decoder (Figure 4) with an Xception-based spatial encoder, an XceptionLSTM temporal encoder and decoder and an MLP as the spatial decoder. The structures of the spatial encoder and the decoder are depicted in Figures 5 and 6, respectively.

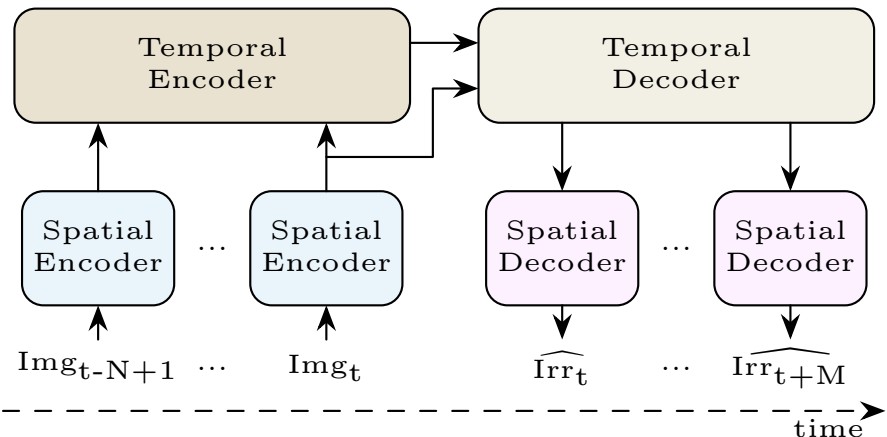

**Figure 4.** Spatio-temporal encoder/decoder breakdown.

The spatial encoder contains 6 layers. The first two layers are the two input XLs that extract data from the input image. Then, the two middle residual layers refine the data. Each of these two residual layers consists of a nested sequential module of two XLs. Each residual layer's output is the sum of the nested module's output and its input. Note here that the number of output tensor channels of the second residual layer is double compared to its input tensor channels. To match the above in the second residual layer, the sum is calculated with the result of a $1 \times 1$ convolution of the input in order to match the number of channels in all nested layers. The nested module can be interpreted as an input corrector that refines the input tensor's data. The last two XLs compress the data to an encoded state. The spatial encoder finally normalizes the encoded state. The spatial decoder involves three linear layers, which are represented by the three last layers in Figure 6. The first three layers in Figure 6 reduce the encoded state produced by the temporal decoder to a fixed sized tensor with adaptive average pooling and flatten it to a vector. The third stage normalizes the vector and forwards it to the 3-layer MLP.

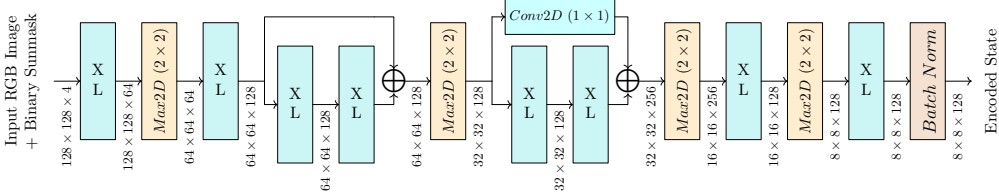

**Figure 5.** Spatial encoder.

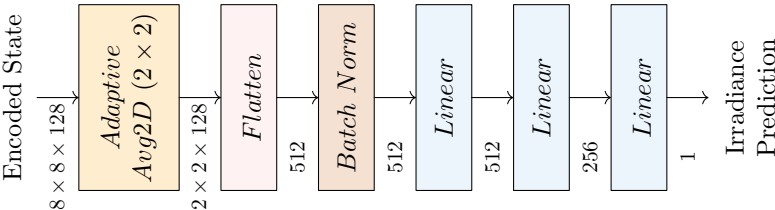

**Figure 6.** Spatial decoder.

All layers are followed by leaky rectified linear unit (LeakyReLU) with a negative slope of 0.1125. ReLU is the output activation function in the inference, but the training scheme uses LeakyReLU with a near-zero negative slope $(10^{-3})$ to allow backpropagation when a negative value is produced in the early stages of training. The proposed model can accept any image of frame size of at least $64 \times 64$ pixels. This work uses frames of size $128 \times 128$. We note here that during the development stages of the proposed model, we studied a variety of models; all these models yielded the best metrics and results with the $128 \times 128$ size frames.

### 2.2. Dataset

The current study has conducted research for datasets appropriate for the model. During the early stages, we considered the Folsom, CA, dataset [26], which was based on a camera that did not provide constant orientation over the time that the dataset was produced; therefore, there was no method to locate the sun systematically. Other datasets include the WSISEG database [27], WILLIAM Meteo Database [28], SKIPP'D [29] and SRRL BMS [30], which are limited with respect to the number of images compared to our needs for training the proposed model and for the target forecasting horizon. Hence, we proceeded by organizing and developing the Archon–Athens, Greece Dataset, a custom dataset for the purposes of the project Archon [9]. The dataset consists of approximately 250 thousand sky images from the All Sky Imager (ASI-16), an automatic full-sky camera system with fisheye lens for a 180° field of view, and GHI measurements of 1 min intervals. The instruments are placed in the rooftop of the Inaccess office (38.04° N, 23.81° E, Sorou Str, Athens, GR) and have gathered data since 25 October 2022. The captured images depict the various weather conditions that occur in the greek capital. Most of the images show partly cloudy or clear sky conditions. There are also plenty of days with precipitation and thin cloud ceilings. Also, the halo phenomenon appears often in images from morning and evening hours. Samples of the dataset are provided in Figure 7.

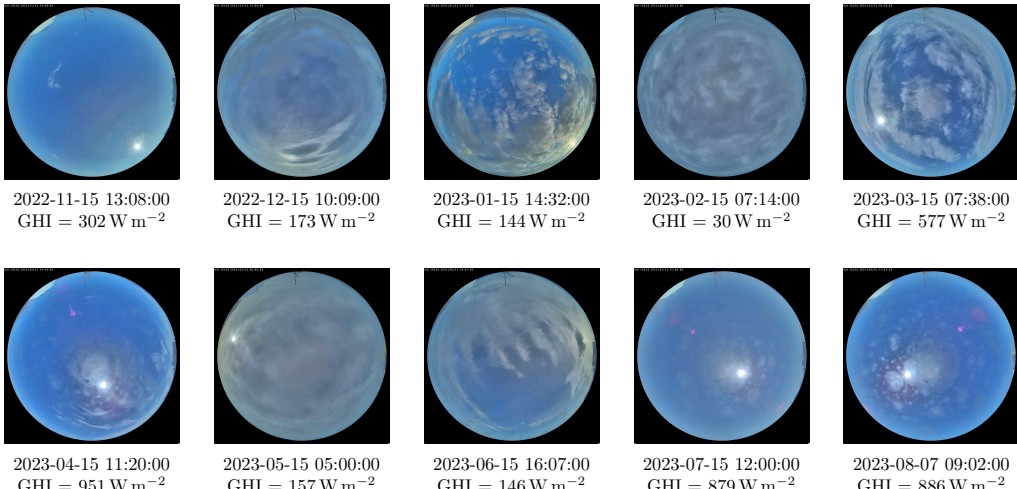

**Figure 7.** The Archon–Athens, Greece Dataset.

We keep all images from January, April and July 2023 as evaluation data and use the rest of the dataset for training and validation with a 90% split. January represents a set of days with bad power yields and frequent weather changes, while April and July represent days with good yields with frequent and infrequent weather changes, respectively. To achieve a more objective validation loss, the training and validation sets are created by splitting the dataset based on the days in a month and then extracting the valid input image and target irradiance sequences from each subset for all months. This guarantees that the model has not processed an image during both training and validation and that the two subsets cover the weather conditions from all the available months. Overall, we used 67%

of the dataset for training, 8% for validation and fine-tuning and 25% for the evaluation presented in Section 3.

### 2.2.1. Input Images

The proposed model accepts as input a sequence of consecutive images captured one frame per minute. All images are first resized from $1536 \times 1536$ RGB images of 8-bit depth to $128 \times 128 \times 3$ tensors of single floats. An extra channel called sun mask [14] highlights the sun disk in an image at clear sky conditions. The introduction of this binary mask is one way to provide the models with useful data related to the image. Such data include the solar azimuth and the solar elevation, and by using this binary mask, they can be easily correlated with the image. It also hints that the highlighted areas are the region in the image expected to correspond to the sky fragments providing the larger fraction of GHI.

### 2.2.2. Output Irradiance

The proposed model outputs irradiance (GHI) values as single floats. These GHI target values are integers and range from one (1) to around 1460 W m$^{-2}$. We have discarded all possible sequences that have sky images captured before sunrise or after sunset.

### 2.3. Metrics

The literature provides a variety of metrics for evaluating solar forecasts, which are envisaged from different perspectives [31]. In this article, we evaluate the results of the tested models with mean bias error (MBE), mean absolute error (MAE), mean absolute percentage error (MAPE), root mean square error (RMSE), and forecast skill (FS) shown in Equations (11)–(15), respectively.

$$\text{Mean bias error:} \qquad \text{MBE} = \frac{1}{N} \sum_{i=1}^{N} (\widehat{y_i} - y_i), \qquad (11)$$

$$\text{Mean absolute error:} \qquad \text{MAE} = \frac{1}{N} \sum_{i=1}^{N} \|\widehat{y_i} - y_i\|, \qquad (12)$$

$$\text{Mean absolute percentage error:} \qquad \text{MAPE} = \frac{1}{N} \sum_{i=1}^{N} \left\| \frac{\widehat{y_i} - y_i}{y_i} \right\|, \qquad (13)$$

$$\text{Root mean square error:} \qquad \text{RMSE} = \sqrt{\frac{1}{N} \sum_{i=1}^{N} (\widehat{y_i} - y_i)^2}, \qquad (14)$$

$$\text{Forecast skill:} \qquad \text{FS} = 1 - \frac{\text{RMSE}}{\text{RMSE}_{\text{pers}}}, \qquad (15)$$

where $N$ is the size of the test dataset, $\widehat{y_i}$ is the forecast and $y_i$ is the target value for a horizon $H$. The MBE highlights whether a model shows bias when forecasting and hence, whether the results tend to consistently under- or overestimate the target value. The MAE and RMSE show the measured deviation of the results in respect to the target values. We can interpret the former as the expected deviation in the lower range of the GHI values, whereas the latter refers to the expected deviation in the upper range of the GHI values. The FS provides a more dataset-independent way to evaluate models [32]. This is accomplished by comparing the models to the Persistence Model, which forecasts that no change will occur to the target value after a horizon. The Persistence Model is a baseline model that often appears in short- and ultra short-term irradiance forecasting solutions, where the forecast horizon ranges from 15 s to 2 min. The MAPE metric indicates the normalized deviation of the forecasts from the target values, which also helps in assessing the models' performance more comprehensively.

### 3. Results

This section presents the evaluation results of the proposed model for the task of short-term irradiance forecasting. Moreover, it presents the results of the study on the performance of models with various temporal encoders/decoders and it compares their results to that of the proposed model. The comparison models include ConvLSTMs, stacked ConvLSTMs, bidirectional ConvLSTMs and their respective depthwise separable (DWSConvLSTM) and Xception versions. The presented benchmark also compares the temporal encoders and decoders that are based on convolutional gated recurrent Units (ConvGRU) [33], an RNN initially intended for spatio-temporal feature learning from videos. We note here that bidirectionality only applies to the temporal encoder, as it is the only module that accepts a sequence as an input, and the results of the forward and backward pass of the input sequence are summed and forwarded to the temporal decoder. All layers of the tested temporal models accept and generate tensors of size $8 \times 8 \times 128$. Table 1 is an overview of the models evaluated in this work.

**Table 1.** Overview of spatio-temporal models' number of parameters and operations and the training time per epoch for an input sequence of five $128 \times 128 \times 4$ images and an output sequence of fifteen irradiance values.

| Temporal Model | Kernel Size | Temporal Encoder/Decoder | | Spatio-Temporal Encoder/Decoder | | Training Time per Epoch (min) |
|---|---|---|---|---|---|---|
| | | Param. | OPs (MAC) | Param. | OPs (MAC) | |
| Spatial Encoder | - | - | - | 833 K | 1.03 G | 3.26 |
| Spatial Decoder | - | - | - | 658 K | 0.66 M | |
| ConvGRU | 3 | 1.79 M | 1.13 G | 3.90 M | 6.27 G | 19.16 |
| | 5 | 4.93 M | 3.15 G | 7.05 M | 8.28 G | 20.54 |
| ConvLSTM | 3 | 2.43 M | 1.51 G | 4.94 M | 6.65 G | 19.33 |
| | 5 | 6.62 M | 4.19 G | 9.13 M | 9.33 G | 20.79 |
| bi-ConvLSTM | 3 | 4.85 M | 1.89 G | 6.80 M | 7.02 G | 20.89 |
| | 5 | 13.2 M | 5.24 G | 15.8 M | 10.4 G | 21.68 |
| Stacked ConvLSTM | 3, 3 | 6.06 M | 3.02 G | 8.57 M | 8.16 G | 21.67 |
| | 3, 5 | 12.3 M | 5.71 G | 14.9 M | 10.8 G | 23.54 |
| | 5, 5 | 16.5 M | 8.39 G | 19.1 M | 13.5 G | 25.45 |
| DWSConvLSTM | 3 | 334 K | 172 M | 2.85 M | 5.31 G | 19.66 |
| | 5 | 342 K | 177 M | 2.85 M | 5.31 G | 19.73 |
| bi-DWSConvLSTM | 3 | 668 K | 215 M | 3.18 M | 5.35 G | 20.13 |
| | 5 | 684 K | 221 M | 3.20 M | 5.36 G | 20.23 |
| Stacked DWSConvLSTM | 3, 3 | 826 K | 343 M | 3.34 M | 5.48 G | 20.64 |
| | 3, 5 | 839 K | 349 M | 3.35 M | 5.49 G | 20.75 |
| | 5, 5 | 847 K | 354 M | 3.36 M | 5.49 G | 20.87 |
| XceptionLSTM | XL | 871 K | 516 M | 3.38 M | 5.65 G | 19.63 |
| bi-XceptionLSTM | XL | 1.74 M | 645 M | 4.25 M | 5.78 G | 19.75 |
| Stacked XceptionLSTM | 2×XL | 2.17 M | 1.03 G | 4.68 M | 6.17 G | 20.79 |

All the models are trained and evaluated in a Linux workstation with an Intel(R) Core(TM) i7-9700K CPU @ 3.60 GHz and a NVIDIA GeForce RTX 3080 GPU. We deploy a Raspberry Pi 4 Model B 8 GB and a Raspberry Pi Zero 2W for time performance tests as devices on the edge, configured as a Linux workstation with a quad core Cortex-A72 (ARM v8) 64-bit SoC @ 1.5 GHz for the former and as a Linux workstation with a quad-core Arm Cortex-A53 64-bit SoC @ 1 GHz for the latter device. We use Python 3.9.13 and Pytorch 2.0.0+cuda11.7 for the development of the evaluated models.

### 3.1. Training Scheme

In order to reduce the total training time of all models that are evaluated in this article, we used transfer learning and partitioned the models' training in two stages, as shown in Figure 8. In the first stage, we trained the spatial encoder and decoder in the training dataset for the problem of irradiance estimation. Specifically, the spatial model accepts an image and estimates the GHI value for this particular image. This stage is common to all the models we tested; therefore, the spatial encoder and decoder were trained only once. We train the second stage's spatio-temporal model using as initial weights: (a) for the spatial encoder and decoder those resulting of the first stage and (b) for the temporal encoder and decoder arbitrary weights. This scheme allows us to test whether the spatial encoder and decoder can effectively forecast GHI values.

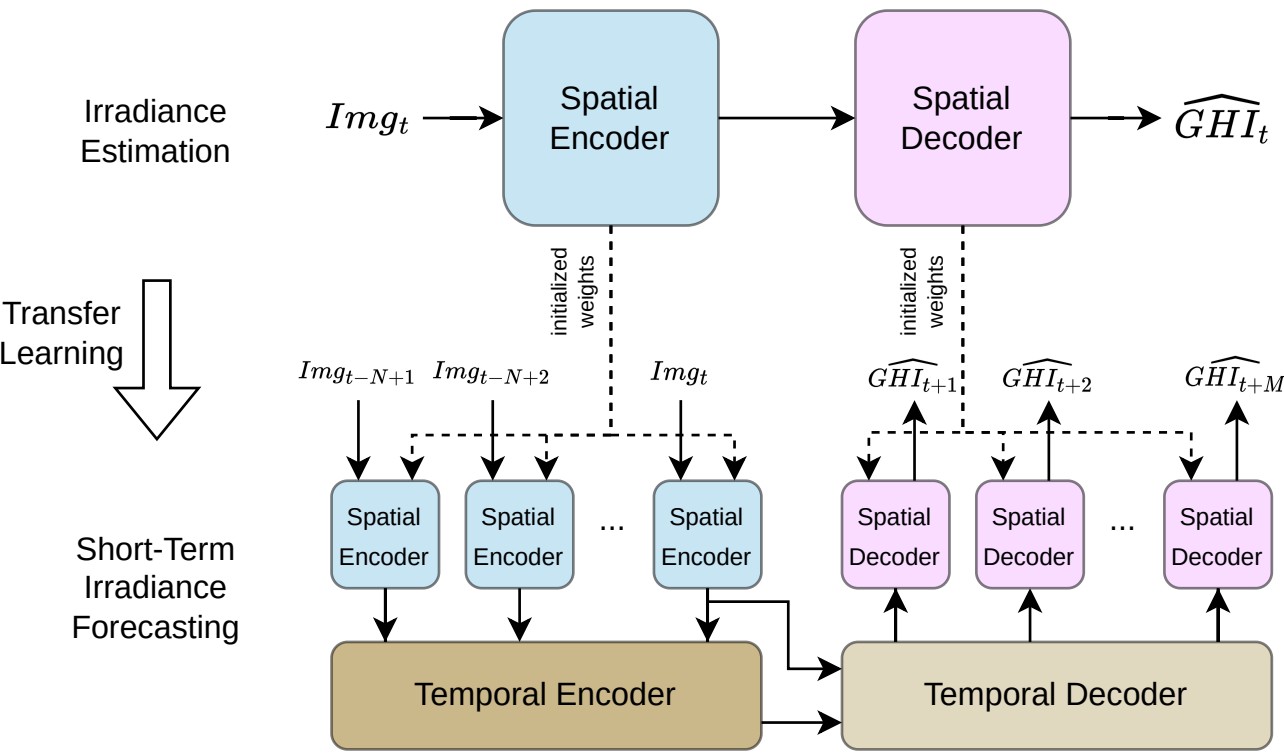

**Figure 8.** Training scheme.

Table 2 lists all the hyperparameters that we examined and chose for the presented benchmark. For the first stage, we used RMSProp with decay of 0.9 and $\epsilon = 1.0$. We used a learning rate of 0.001 decaying every epoch using an exponential rate of 0.94 and the MSELoss as the criterion for calculating the loss. The spatial model was trained for 29 epochs and achieved an RMSE of 35.3 W m$^{-2}$ for the solar irradiance estimation. For the second stage, we used the same scheduler, optimizer and loss function with an initial learning rate of $5 \times 10^{-5}$. With this training scheme we were able to reduce the total epochs from 30 to just 6 epochs for the spatio-temporal models. Moreover, all the models achieved better metrics when they were trained with this scheme compared to training the whole spatio-temporal model without any transfer learning.

**Table 2.** The hyperparameters that were examined and chosen for the presented benchmark.

| Hyperparameter | Tested Options | Final |
|---|---|---|
| Input Sequence Length | {5, 10, 15} | 5 |
| Output Sequence Length | {5, 10, 15} | 15 |
| Image Frame Size | {64, 128, 256} | 128 |
| Concatenate Sunmask | {True, False} | True |
| Removed Foreign Objects | {True, False} | True |
| Encoded State's Channels | {16, 32, 64, 128, 256} | 128 |
| Optimizer | {Adam, RMSProp} | RMSProp |
| Scheduler | ReduceOnPlateau, Exponential, Step | Exponential |
| Learning Rate | $5 \times 10^{-2}$, $10^{-3}$, $5 \times 10^{-4}$, $10^{-4}$, $5 \times 10^{-5}$, $10^{-5}$ | 1st Stage: $10^{-3}$ 2nd Stage: $5 \times 10^{-5}$ |
| Loss Function | L1, SmoothL1 Huber, MSE | MSE Loss |
| Batch Size | {8, 12, 16, 20, 24} | 16 |

*3.2. Model Evaluation*

Table 1 presents the implementation details of the models evaluated in this work: the ConvLSTM, the DWSConvLSTM and the XceptionLSTM-based models in their single-cell, bidirectional and double-stacked versions and the ConvGRU-based models. Table 3 presents the evaluation results of the metrics for the tested models. All the models considered for this comparison infer a sequence of five images with a horizon of 1 min and output 15 GHI values that correspond from 1 to 15-min forecasts. Note here that the XceptionLSTM cell in the three versions we examined prevails with respect to the RMSE and FS scores. Specifically, the single XceptionLSTM cell scores 68.8 W m$^{-2}$, 94.6 W m$^{-2}$ and 116 W m$^{-2}$ RMSE for the horizons of 1, 5, 15 min with a mean score of 99.8 W m$^{-2}$ and the double-stacked XceptionLSTM scores 69.2 W m$^{-2}$, 94.4 W m$^{-2}$ and 115.8 W m$^{-2}$ RMSE with a mean score of 99.8 W m$^{-2}$. Moreover, the proposed cell in its bidirectional form scores 68.7 W m$^{-2}$, 94.5 W m$^{-2}$ and 117.3 W m$^{-2}$ in the RMSE metric for the same horizons and reports a mean score of 100.3 W m$^{-2}$. The bidirectional XceptionLSTM is the best-performing model based on the MAPE metric, achieving a mean score of 24.5% across the examined horizons. Furthermore, the XceptionLSTM cell achieves low MAE scores for the 1 min horizon, but the stacked DWSConvLSTM cells report lower MAE metrics in all the other horizons.

The MBE metric reports no noticeable bias for ConvLSTM and XceptionLSTM-based spatio-temporal models, which means that these two kinds of models have no tendency to over- or underestimate forecasts. On the other hand, most DWSConvLSTM-based models tend to systematically underestimate the forecasts of the first few horizons. The same tendency appears in the bidirectional versions of all models. In contrast to the above, the ConvGRU based models tend to overestimate forecasts. Moreover, the models with bidirectional temporal encoders appear to forecast more accurately for greater horizons when compared to their unidirectional counterparts, but less accurately when one more layer is added to the unidirectional models. As we can conclude by the metrics of Table 3, the increased accuracy of the models with stacked temporal encoders and decoders seems to be more intense in the cells that use depthwise operations. In addition, given that all the benchmarked models score RMSE and MAE values within 4.6 W m$^{-2}$ from one another, we conclude that the structure of the temporal encoder and decoder does not significantly change the reported metric scores.

**Table 3.** Evaluation results of the models in Table 1 for the horizons of 1, 5, 15 min and the average for the first 15 min. The best models for each metric and horizon are **bolded**.

| Model | Kernel Size | MBE (W m$^{-2}$) | | | | MAE (W m$^{-2}$) | | | | MAPE (%) | |
|---|---|---|---|---|---|---|---|---|---|---|---|
| | | 1 min | 5 min | 15 min | Mean | 1 min | 5 min | 15 min | Mean | 1 min | 5 min |
| Persistence | – | **<0.025** | −0.0841 | 0.1795 | **<0.001** | **20.61** | **44.1** | 74.8 | 52.9 | **6.21** | **15.89** |
| ConvGRU | 3 | −3.19 | 9.63 | 9.09 | 8.83 | 34.6 | 47.2 | 59.6 | 50.4 | 24.66 | 32.5 |
| | 5 | −1.508 | 7.86 | 8.17 | 7.52 | 33.0 | 46.2 | 59.0 | 49.7 | 22.72 | 29.34 |
| ConvLSTM | 3 | −0.686 | 5.68 | 3.82 | 3.60 | 33.1 | 47.0 | 59.6 | 49.9 | 21.41 | 27.73 |
| | 5 | 2.080 | 1.898 | −1.321 | 0.1465 | 33.0 | 47.1 | 60.9 | 50.8 | 21.73 | 27.87 |
| bi-ConvLSTM | 3 | 3.02 | 2.520 | **<0.025** | 0.1741 | 34.1 | 49.2 | 63.3 | 53.5 | 19.89 | 26.69 |
| | 5 | −4.24 | −5.07 | −7.30 | −6.10 | 34.3 | 47.4 | 60.0 | 50.8 | 19.91 | 24.87 |
| Stacked ConvLSTM | 3, 3 | 3.10 | 3.26 | 1.304 | 1.741 | 33.5 | 47.3 | 59.5 | 50.3 | 22.08 | 29.31 |
| | 3, 5 | 0.2465 | 0.313 | −4.12 | −1.500 | 33.0 | 47.0 | 60.0 | 50.4 | 21.49 | 27.68 |
| | 5, 5 | −2.170 | −1.474 | −4.08 | −2.824 | 35.0 | 47.7 | 59.9 | 51.0 | 19.76 | 26.35 |
| DWSConvLSTM | 3 | −14.13 | −5.37 | −2.234 | −4.49 | 35.5 | 46.6 | 59.8 | 50.3 | 22.11 | 27.69 |
| | 5 | −11.49 | −5.97 | −3.01 | −4.75 | 34.8 | 46.3 | 59.0 | 49.8 | 21.56 | 26.20 |
| bi-DWSConvLSTM | 3 | −15.90 | −5.63 | 0.988 | −3.94 | 37.0 | 46.0 | 58.5 | 49.5 | 22.71 | 26.88 |
| | 5 | 3.54 | 3.81 | −4.48 | 1.335 | 34.5 | 46.6 | 60.1 | 50.1 | 23.07 | 28.81 |
| Stacked DWSConvLSTM | 3, 3 | −6.38 | 0.678 | 2.481 | 1.022 | 35.0 | 46.4 | **58.5** | 49.6 | 23.14 | 29.92 |
| | 3, 5 | −8.51 | −3.71 | 1.160 | −1.831 | 34.7 | 45.4 | 58.3 | **49.0** | 20.60 | 26.18 |
| | 5, 5 | 1.994 | 0.452 | −6.07 | −1.518 | 34.5 | 46.7 | 59.6 | 50.1 | 23.01 | 28.48 |
| XceptionLSTM | XL | −3.01 | 1.139 | −4.87 | −0.924 | 32.5 | 46.1 | 59.9 | 49.7 | 20.46 | 24.43 |
| bi-XceptionLSTM | XL | −7.03 | −2.992 | −8.16 | −4.52 | 32.7 | 45.7 | 59.0 | 49.2 | 19.57 | 22.98 |
| Stacked XceptionLSTM | 2×XL | −2.650 | −0.963 | −3.35 | −1.657 | 33.0 | 45.8 | 60.4 | 49.9 | 19.38 | 23.27 |

| Model | Kernel Size | RMSE (W m$^{-2}$) | | | | FS (%) | | | | MAPE (%) | |
|---|---|---|---|---|---|---|---|---|---|---|---|
| | | 1 min | 5 min | 15 min | Mean | 1 min | 5 min | 15 min | Mean | 15 min | Mean |
| Persistence | – | 75.2 | 113.3 | 146.6 | 122.4 | – | – | – | – | 38.5 | 30.7 |
| ConvGRU | 3 | 69.4 | 94.8 | 113.1 | 99.9 | 7.70 | 16.37 | 22.88 | 17.77 | 38.5 | 30.7 |
| | 5 | 68.3 | 95.4 | 114.6 | 100.0 | 9.12 | 15.77 | 21.83 | 17.74 | 36.8 | 31.4 |
| ConvLSTM | 3 | 69.5 | 95.5 | 114.5 | 100.0 | 7.50 | 15.69 | 21.92 | 17.67 | 38.5 | 30.7 |
| | 5 | 70.0 | 98.2 | 116.2 | 102.1 | 6.94 | 13.34 | 20.75 | 15.93 | 37.2 | 30.5 |
| bi-ConvLSTM | 3 | 70.2 | 96.5 | 115.5 | 101.3 | 6.61 | 14.85 | 21.22 | 16.58 | 33.2 | 28.38 |
| | 5 | 70.1 | 95.9 | 115.6 | 100.9 | 6.79 | 15.36 | 21.13 | 16.94 | 31.1 | 26.53 |
| Stacked ConvLSTM | 3, 3 | 69.7 | 96.3 | 115.1 | 100.7 | 7.27 | 15.06 | 21.47 | 17.09 | 36.7 | 31.0 |
| | 3, 5 | 70.2 | 97.8 | 116.5 | 102.2 | 6.67 | 13.66 | 20.52 | 15.87 | 34.3 | 29.43 |
| | 5, 5 | 70.7 | 96.5 | 115.0 | 101.1 | 5.93 | 14.88 | 21.56 | 16.72 | 31.9 | 27.64 |
| DWSConvLSTM | 3 | 72.2 | 96.7 | 117.0 | 101.7 | 3.96 | 14.65 | 20.17 | 16.15 | 37.6 | 30.5 |
| | 5 | 71.4 | 95.2 | 115.2 | 100.3 | 5.02 | 15.97 | 21.39 | 17.31 | 34.9 | 28.99 |
| bi-DWSConvLSTM | 3 | 72.3 | 95.0 | 115.1 | 100.0 | 3.79 | 16.14 | 21.50 | 17.51 | 35.7 | 29.55 |
| | 5 | 71.0 | 95.7 | 114.8 | 100.2 | 5.57 | 15.57 | 21.71 | 17.37 | 38.8 | 31.6 |
| Stacked DWSConvLSTM | 3, 3 | 73.3 | 95.9 | **113.7** | 100.3 | 2.51 | 15.38 | **22.43** | 17.16 | 36.7 | 31.8 |
| | 3, 5 | 72.1 | 95.2 | 114.3 | 100.0 | 4.09 | 16.01 | 22.01 | 17.55 | 35.2 | 28.72 |
| | 5, 5 | 71.1 | 95.5 | 114.4 | 100.2 | 5.41 | 15.71 | 21.99 | 17.42 | 39.4 | 31.4 |
| XceptionLSTM | XL | 68.8 | 94.6 | 116.0 | **99.8** | 8.53 | 16.52 | 20.88 | **17.85** | 32.3 | 27.09 |
| bi-XceptionLSTM | XL | **68.7** | 94.5 | 117.3 | 100.3 | **8.52** | 16.57 | 20.01 | 17.52 | **28.91** | **24.50** |
| Stacked XceptionLSTM | 2×XL | 69.2 | **94.4** | 115.8 | 99.8 | 7.94 | **16.69** | 21.04 | 17.85 | 31.3 | 25.66 |

*3.3. Timing Reports*

Figure 9 is a diagram of the mean execution time of the evaluated models on the edge devices. The proposed model, in its three benchmarked versions, executes as a single cell in 2.69 s and 7.54 s, as a bidirectional cell in 2.91 s and 7.99 s and as a double-stacked cell 3.31 s and 8.78 s for the Raspberry Pi 4 Model B and the Raspberry Pi Zero 2W. We notice that the inference time of all models are within 10 % of the slowest recorded time on both edge devices, which corresponds to the model that includes stacked XceptionLSTMs in its temporal encoder and decoder. This is because a major fraction of the complexity derives from the repeated execution of the spatial encoder and decoder, that is, once per element of the input and output sequences. The graph also shows the expected behavior for the models that are based on the same LSTM cells, a fact deducing that the model with less parameters executes faster. That behaviour is not true for cells of different structure; despite the great difference in the amount of parameters and total operations, XceptionLSTM cells appear to have comparable execution times with ConvLSTM cells due to the optimizations that the Pytorch library performs in convolutions. We believe that the reason behind this is that the operation of concatenating the channels of the results of all the nested depthwise operations in an XL to form a single tensor for the pointwise convolution to process causes the reported execution time overhead. One way to cope with this is to avoid concatenation by executing the pointwise operation first, then splitting the intermediate tensor and finally, forwarding the chunks to the nested depthwise operations. These modifications improve the reported execution times but they result in degraded metrics. During the evaluation of the inference times, both devices reported a constant power consumption measured at 5.1 W for the Raspberry Pi 4 Model B and 0.7 W for the Raspberry Pi Zero 2W.

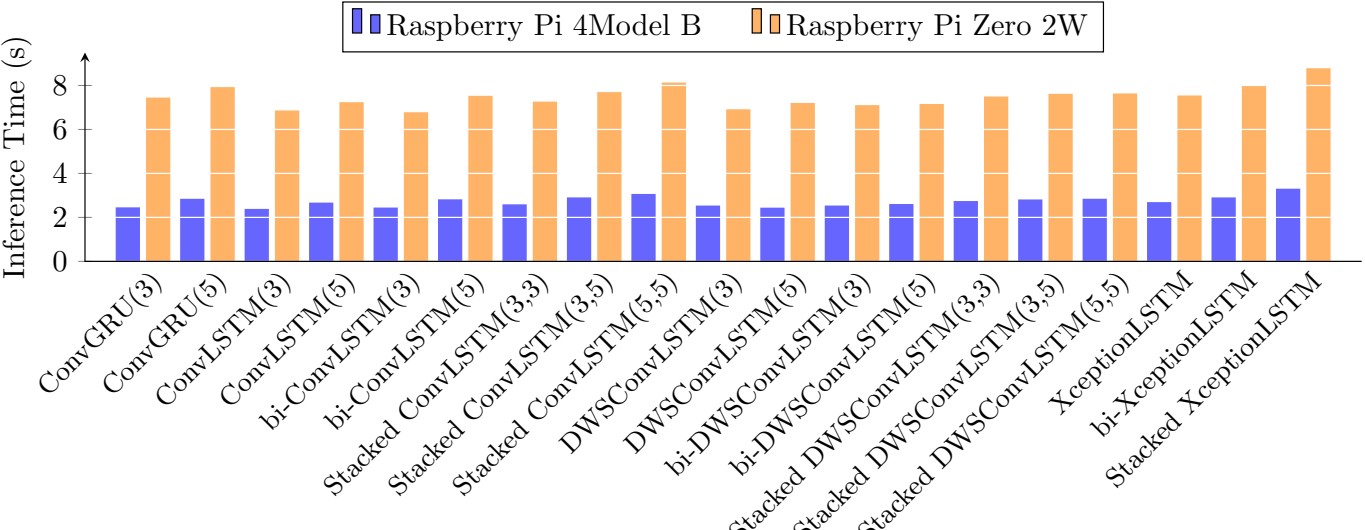

**Figure 9.** Timing reports for inference on low-cost, edge computing devices.

## 4. Discussion

Researchers and engineers in the renewable energy field are keen for solutions to the short-term irradiance forecasting problem [31]. Especially in the last two decades, they focus on computer vision and ML-based systems, which often include image processing for satellite imagery [34–37]. Given that the satellite images cover a vast area of the Earth's land, measuring the GHI in different areas of an image may provide significantly different values. The alternative is the ground-based imagery [38–40], which clearly depicts the current weather conditions in the area of interest with a notable application example the case of large PV parks. The PV park controllers use multiple ground-based sensors and they can yield more accurate results [41].

As Ziyabari et al. [10] suggest, researchers often consider spatio-temporal architectures as a solid base for their models because the dimensionality of the input data does not constrain considerably the final structure of their models. This holds whether the input data is multiple time-series of environmental measurements from multiple sensors that are spread in a wide area or, as in this article, a time-series of images from a single sky camera. ConvLSTM-based solutions are reported in image regression related techniques that target irradiance forecasting [42–44]. This is because CLs are effective in modelling the complex dynamics of the environmental variables, such as the cloud and the wind movement. Moreover, as the name suggests, ConvLSTMs can capture the long-term evolution of the irradiance values. More accurately, ConvLSTMs excel in modeling the long-term dependences of the target data and extract the correlation among the input data [40]. It is quite common for researchers to utilize image segmentation for cloud cover estimations as a means to enhance the results of ML-based forecasting models [45,46].

Zhang et al. [47] compare the results of MLP, CNN and LSTM models that are trained to predict PV power differences by using PV power data and sky images. They conclude that a hybrid model using both PV power data and images has a better-balanced performance across different types of weather conditions. Sun et al. [48] present the SUNSET, a deep CNN architecture that accepts an image sequence and other data produced by the PV park and it outputs PV power and clear sky index (CSI) predictions. The input image sequence is in the form of a single hyperspectral image. Ajith et al. [49] developed a multi-modal fusion network for ultra-short irradiance forecasting using infrared images and past irradiance data. They explain that infrared images of the sky can better capture the cloud dynamics in the small horizon of 15 s. Kumari et al. [40] discuss the advantages and drawbacks of using LSTMs, gated reccurrent units (GRU), CNNs, deep belief networks (DBN), RNNs and hybrid artificial neural networks (ANN) for solar irradiance forecasting. Basmile et al. [50] review and compare eight different AI models for horizons of a minute, an hour and for daily average forecasts of GHI, DHI and DNI values. Nie et al. [44] explore training tactics for heterogeneous datasets and how transfer learning contributes to reducing the training effort and improving the results of a model. Lyu et al. [51] use deep reinforcement learning (DRL) in order to dynamically change between optimal features of a model by recognising weather patterns. We note here that the FS reported in the bibliography ranges from $-2.4 - 33.2\%$ for the lower range to $14.4 - 25.2\%$ for the upper range of the examined horizons [14,43,52,53].

The proposed model targets implementations for short-term irradiance forecasting on low-power devices. In this article, we presented a benchmarking of known ConvLSTM based spatio-temporal models [10,19,40] for the latter task with the evaluation of the models in terms of metric scores and execution times. The proposed XceptionLSTM cell and spatio-temporal model show notable performance for horizons over 10 min and improved forecasting skills for smaller horizons. We noticed that, when taking into account the evaluation results of other related works [43,54], the proposed model exhibit a lower drop of forecasting skill as the horizon increases. This means that our model constitutes a very attractive solution for short-term irradiance forecasting; moreover, it can be integrated in SG systems that are based on ultra short-term forecasts. Furthermore, the proposed RNN is significantly lightweight when compared to traditional RNNs and models from the bibliography [43,44], as it requires half of the memory that the weights of the ConvLSTM-based spatio-temporal models need.

Focusing on the edge devices, the proposed model is optimized for inference on low-power devices and can process up to 22.27 sequences per minute in the low power device Raspberry Pi 4 Model B and up to 7.81 sequences per minute in the ultra low-power device Raspberry Pi Zero 2W. Hence, less powerful IoT devices suffice for the execution of forecasting tasks that were once considered power intensive and computationally demanding. Considering that the number of PV parks added to the power grids constantly increases and that low-end devices are easy to maintain and cost-effective, the proposed model offers a very tempting alternative to the high performance and high cost controllers.

We note here that, on one hand, the real-time execution of the inference computations on the edge will impose limitations on the NN design and consequently on the number of respective NN layers. On the other hand, the models in the bibliography that consider mainly the improvement of feature extraction and not the execution efficiency consist of dozens of sequentially connected layers [44,52,53]. Compared to the later works, the proposed model has limited NN layers but it achieves competitive results and conforms to the specified execution times. An addition to the above cost and execution related considerations in the proposed NN is the exclusion of the injection of numeric data in the final NN layers. This injection [40,44,47–49,51,52] gives more accurate forecasts but increases the overall complexity. Finally, one of the key factors of this research is the response time of the PV park controller. Although the presented IFS is statically configured, the controller must be able to dynamically configure the IFS in order to adjust to the environment and satisfy the time constraints. The controller has to choose the optimal input and output sequence lengths and the horizon for improved PV park management. Therefore, the proposed solution has to be configurable and also maintain a certain level of accuracy across the allowed configurations.

## 5. Conclusions

The current paper introduced a novel irradiance forecasting model that is efficient with respect to the computational complexity and specifically designed for edge computing devices. It also introduced the basis for the design of the forecast model, which is an innovative RNN called XceptionLSTM. The advantages of the proposed model lie in its reduced computational complexity, the achievement of Forecast Skill of 16.57% for a horizon of 5 min when compared to the Persistence Model and finally, the execution time results on known edge computing devices. The results, which are accomplished on a Raspberry Pi 4 Model B 8 GB and on a Raspberry Pi Zero 2W, validate the real-time performance of the model. The future work will first include further optimizations of the XL that will allow for real-time deployment for under a minute horizons. The second very interesting target includes the combination of the XLs with other RNNs, as well as experiments on NN quantization for efficient mapping on very large-scale integration (VLSI) and edge-oriented field programmable gate arrays (FPGA).

**Author Contributions:** Conceptualization, G.V., C.V. and D.R.; methodology, G.V., C.V., A.T. and D.R.; software, G.V., C.V. and A.T.; validation, T.A. and P.G.; formal analysis, G.V. and D.R.; investigation, C.V. and G.K.; resources, D.R. and G.K.; writing—original draft preparation, G.V. and C.V.; writing—review and editing, G.V., C.V., A.T., T.A. and D.R.; supervision, G.K. and D.R.; project administration, G.K.; funding acquisition, G.K. All authors have read and agreed to the published version of the manuscript.

**Funding:** This research has been co-financed by the European Regional Development Fund of the European Union and Greek national funds through the Operational Program Competitiveness, Entrepreneurship and Innovation, under the call RESEARCH—CREATE—INNOVATE (project name "ARCHON" and project code: T2EDK-00864).

**Data Availability Statement:** The *Archon–Athens, Greece Dataset* targeting short-term irradiance forcasting is first introduced in the current work and will soon be available at the project's site http://archonproject.eu/english.html (accessed on 29 August 2023).

**Acknowledgments:** The authors would like to thank Vasilios Kalekis for his work in the early models that led to this study.

**Conflicts of Interest:** Authors Tzouma Amrou, Georgios Konstantoulakis and Panagiotis Golemis are employed by the company Inaccess Networks. The remaining authors declare that the research was conducted in the absence of any commercial or financial relationships that could be construed as a potential conflict of interest.

## Abbreviations

The following abbreviations are used in this manuscript:

| | |
|---|---|
| AI | Artificial intelligence |
| ANN | Artificial neural network |
| CC | Cloud cover |
| CL | Convolutional layer |
| CNN | Convolutional neural network |
| ConvLSTM | Convolutional long short-term memory |
| ConvGRU | Convolutional gated recurrent unit |
| CPU | Central processing unit |
| CSI | Clear sky index |
| DBN | Deep belief network |
| DL | Deep learning |
| DHI | Diffuse horizontal irradiance |
| DNI | Direct normal irradiance |
| DRL | Deep reinforcement learning |
| DWSC | Depthwise separable convolution |
| DWSConvLSTM | Depthwise separable convolutional long short-term memory |
| FC-LSTM | Fully connected long short-term memory |
| FPGA | Field programmable gate array |
| FS | Forecast skill |
| GHI | Global horizontal irradiance |
| GPU | Graphics processing unit |
| GRU | Gated recurrent unit |
| IFS | Irradiance forecasting system |
| IoT | Internet of Things |
| LeakyReLU | Leaky rectified linear unit |
| LSTM | Long short-term memory |
| MAE | Mean absolute error |
| MBE | Mean bias error |
| ML | Machine learning |
| MLP | Multilayer perceptron |
| NLP | Natural language processing |
| NN | Neural network |
| PV | Photovoltaic |
| ReLU | Rectified linear unit |
| RES | Renewable energy source |
| RGB | Red green blue |
| RMSE | Root mean square error |
| RNN | Recurrent neural network |
| Seq2Seq | Sequence-to-sequence |
| SG | Smart grid |
| SoC | System-on-chip |
| VLSI | Very large-scale integration |
| XceptionLSTM | Xception long short-term memory |
| XL | Xception layer |

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
