# Peer review of "Neural Network-Based Solar Irradiance Forecast for Edge Computing Devices"

_information, doi:10.3390/info14110617_

Round 1

Reviewer 1 Report

Comments and Suggestions for Authors

see attachment

Author Response

Thank you for useful comments. Please see the attachment.

Reviewer 2 Report

Comments and Suggestions for Authors

The paper discusses the design and development of a system for short-term irradiance forecasting for photovoltaic (PV) parks. The main goal of this system is to enhance the operation of PV parks and ensure the stability of the electricity grid. The paper introduces the Xception Long Short-Neural Network as a method for achieving accurate irradiance forecasting. The paper is very good in terms of scientific depth. I have some points to improve the manuscript:

1. While the proposed model is already designed to be lightweight, there may be room for further optimization to reduce computational complexity and improve efficiency.

2: The model could benefit from the inclusion of additional relevant features that may contribute to more accurate forecasting. This could involve integrating data from other sensors or sources to provide a more comprehensive input.

3. To ensure the generalizability of the proposed model, it would be beneficial to evaluate its performance on diverse datasets from different geographical locations and weather conditions. This would provide a more comprehensive understanding of its capabilities and limitations.

4: Experimenting with different hyperparameter settings could potentially enhance the performance of the model. This includes parameters such as learning rate, batch size, and regularization techniques.

5: While the evaluation in the context mentions metrics like mean bias error (MBE), mean absolute error (MAE), root mean square error (RMSE), and forecast skill (FS), it may be beneficial to include other metrics that provide a more comprehensive assessment of the model's performance. For example, metrics like precision, recall, and F1 score can provide insights into the model's ability to correctly classify different solar irradiance levels.

6: The future direction of research should be mentioned in the conclusion section.

Comments on the Quality of English Language

Minor editing of English language required

Author Response

Thank you for your useful comments. Please see the attachment.

Reviewer 3 Report

Comments and Suggestions for Authors

The research aimed to improve solution to the GHI forecasting problem executable on edge devices. The research exploited image sequence regression techniques to introduce the Xception Long Short-Term Memory (XceptionLSTM), a recurrent NN (RNN) for Image Sequence parsing and generation, and a model for a complete solution to the GHI forecasting. The proposed model converges faster and requires less memory to infer data, making it ideal for executing inference on edge devices.

The manuscript is well presented and follows the proper methodology. However, following concerns should be considered revising the manuscript.

a. Describe the motivation clearly.

b. Compare the proposed model with the existing works

c. What is the novelty of this research?

d. Compare the results with existing related works.

e. What are the limitations of the research?

Comments on the Quality of English Language

English needs improvement. 

Author Response

(The authors gave the same response as above.)

Reviewer 4 Report

Comments and Suggestions for Authors

Dear Authors 

The paper entitled “Neural Network based Irradiance Forecast  for Edge Computing Devices” reports a study regarding the possibility to forecast at short term the irradiance, to implement a forecasting system on portable and low-energy requiring devices associated to the management of photovoltaic parks. The idea is valid, and the software implementation is deeply and detailly described. Some issues aroused anyway, that require to be solved in order to make the publication possible.

The first concern regards the used portable devices. The Authors tested their application on a Linux Workstation (described in lines 239 and following). Then, in the Manuscript, two portable devices are introduces (Raspberry Pi 4 Model B and Raspberry Pi Zero 2W). If  the Reviewer understood correctly, the main idea is to implement the described Neural Network on some portable devices which do not require high energy and are less expensive than a complete workstation. And yet this main idea is not clearly described anywhere. This concept should be clarified, extended and clearly stated.

Related to the previous point, on line 17 Authors should provide some references to show works were AI has been integrated on smart systems.

On line 60 the Archon project has been nominated: some explanation of the project has to be reported.

On section 2.2 the usage of dataset is explained, but it’s not clear:

-        the 90% split has to be clarified: what does it mean? 90% training and 10% test? If yes: why so unbalanced and not the classic 70%-30% or 80%-20%?

-        Last lines are not clear (200 – 202) are not clear, please specify the splitting method

Section 3.3 regards an interesting aspect, and yet has some serious flaws. First of all, Figure 9 does clearly states haw Raspberry Pi4 Model B requires half of the time of Raspberry Pi Zero, but no numerical values are reported in the text. Some statistic has to be reported, or the graphic of the Figure is not useful.

The comparison between LSTMs has also to be numerically validated, since like that is not understandable the difference between them.

Discussion section is filled with references and comment on works of other groups, but poor on considerations regarding the implemented work. The Reviewer suggest to expand this section and reduce the description of other works.

Conclusions section is too short. Authors have to expand it or, alternatively, join it to the previous section.

Comments on the Quality of English Language

Minor English editing must be carried out

Author Response

(The authors gave the same response as above.)

Round 2

Reviewer 3 Report

Comments and Suggestions for Authors

Thanks for revising the manuscript. It is now in good shape. Good luck!@

Comments on the Quality of English Language

Fine

Reviewer 4 Report

Comments and Suggestions for Authors

The Reviewer want to thanks the Author for replying to the proposed comments. The Manuscript can now - in the Reviewer's opinion - be published.

Best regards